# Neutrophil-to-Lymphocyte Ratio (NLR) Is a Promising Predictor of Mortality and Admission to Intensive Care Unit of COVID-19 Patients

**DOI:** 10.3390/jcm11082235

**Published:** 2022-04-16

**Authors:** Matteo Regolo, Mauro Vaccaro, Alessandra Sorce, Benedetta Stancanelli, Michele Colaci, Giuseppe Natoli, Mario Russo, Innocenza Alessandria, Massimo Motta, Nicola Santangelo, Letizia Fiorito, Ornella Giarrusso, Federica Giangreco, Andrea Arena, Paola Noto, Claudio Ciampi, Giuseppe Carpinteri, Lorenzo Malatino

**Affiliations:** 1Department of Clinical and Experimental Medicine, University of Catania, 95123 Catania, Italy; matteo.regolo.94@gmail.com (M.R.); michele.colaci@unict.it (M.C.); alessandriainnocenza@outlook.it (I.A.); massimo.motta@unict.it (M.M.); letizia.fio90@gmail.com (L.F.); giarrusso.ornella@gmail.com (O.G.); federicagiangreco90@gmail.com (F.G.); claudiociampi@yahoo.it (C.C.); 2Department of Emergency Medicine, San Marco-Polyclinic Academic Hospital, 95124 Catania, Italy; maurovaccaro90@gmail.com (M.V.); p.noto@hotmail.it (P.N.); gicarpinteri@gmail.com (G.C.); 3Department of Health Promotion Sciences, Maternal and Infant Care, Internal Medicine and Medical Specialties, “G. D’Alessandro” (PROMISE), Unit of Nephrology and Hypertension, European Society of Hypertension Excellence Centre, University of Palermo, 90133 Palermo, Italy; avinouros85@gmail.com; 4Academic Unit of Internal Medicine, Cannizzaro Hospital, 95100 Catania, Italy; benedetta.stancanelli@virgilio.it (B.S.); giuseppenatoli@hotmail.it (G.N.); russom32@gmail.com (M.R.); nicola.santangelo@libero.it (N.S.); andrea.arena91@gmail.com (A.A.)

**Keywords:** NLR, PLR, SARS-CoV2, neutrophil-to-lymphocyte ratio, CRP, inflammation, mortality, prognostic biomarkers, intensive care unit, ICU

## Abstract

The neutrophil-to-lymphocyte ratio (NLR) is an inflammatory marker predicting the prognosis of several diseases. We aimed to assess its role as a predictor of mortality or admission to the intensive care unit in COVID-19 patients. We retrospectively evaluated a cohort of 411 patients with COVID-19 infection. The neutrophil-to-lymphocyte ratio (NLR), platelet-to-lymphocyte ratio (PLR), and C-reactive protein (CRP) of patients with COVID-19 were compared. The median age of our sample was 72 years (interquartile range: 70–75); 237 were males. Hypertension, diabetes and ischemic heart disease were the most common comorbidities. The study population was subdivided into three groups according to NLR tertiles. Third-tertile patients were older, showing significantly higher levels of inflammatory markers; 133 patients (32%) died during hospitalization, 81 of whom belonged to the third tertile; 79 patients (19%) were admitted to ICU. NLR showed the largest area under the curve (0.772), with the highest specificity (71.9%) and sensitivity (72.9%), whereas CRP showed lower sensitivity (60.2%) but slightly higher specificity (72.3%). Comparisons between NLR and CRP ROC curves were significantly different (*p* = 0.0173). Cox regression models showed that the association between NLR and death was not weakened after adjustment for confounders. Comparisons of ROC curves showed no significant differences between NLR, PLR, and CRP. Cox regression analysis showed that NLR predicted the risk of admission to ICU independently of demographic characteristics and comorbidities (HR: 3.9597, *p* < 0.0001). These findings provide evidence that NLR is an independent predictor of mortality and a worse outcome in COVID-19 patients and may help identify high-risk individuals with COVID-19 infection at admission.

## 1. Introduction

Symptoms of COVID-19 are non-specific, and clinical manifestations can range from no symptoms to severe pneumonia and acute respiratory distress. Any age is susceptible to SARS-CoV-2 infection, but elderly patients and those with pre-existing comorbidities are particularly prone to severe illness [1,2].

Inflammation plays a key role in the development of severe COVID-19 disease. Severe inflammatory response during COVID-19 results in a weakened adaptive immune function, thereby unbalancing the immune system. Early identification of risk factors for patients with greater clinical severity is vital to allow appropriate supportive care or access to intensive treatments if needed [3].

Given the wide spectrum of possible clinical presentations and the potential variability during disease evolution, the recognition of the hyper-inflammatory state is the first step in the characterization of patients, in order to treat them in the most appropriate way, according to the stage of the disease.

Circulating biomarkers of inflammation play a predominant role in this regard. A simple serum test would be required to assess the systemic involvement and to help predict the outcome of patients [4].

Several factors can be used in the evaluation of inflammatory status. Apart from well-recognized biomarkers of inflammation, such as C-reactive protein (CRP), procalcitonin and interleukin-6 (IL-6), neutrophil-to-lymphocyte ratio (NLR) and platelet-to-lymphocyte ratio (PLR) have been recently proposed to predict the severity of COVID-19 disease [5]. In particular, NLR is a simple ratio obtained by dividing the absolute neutrophil and lymphocyte counts from the leukocyte formula. We have recently shown that NLR can predict the prognosis of patients with community-acquired pneumonia (CAP) [6], urothelial cancer treated with immunotherapy [7], as well the presence of carotid atherosclerotic plaques in older patients [8].

Of note, neutrophils and lymphocytes are closely related to the pathophysiology of COVID-19. Current evidence suggests that neutrophilia and lymphocytopenia both reflect the natural physiologic response of circulating leukocytes to pathophysiological processes following stress, trauma, major surgery, bacteremia states, systemic inflammation, as well as systemic inflammatory response syndrome (SIRS) and sepsis [9,10,11].

Thus, from the pathophysiologic viewpoint, the cytokine storm promotes the massive recruitment of neutrophils through activation and chemotaxis and determines a significant reduction in the number of circulating lymphocytes due to depletion, consumption, and negative counter-regulation [12].

In the present study, we aimed to assess the prognostic value of NLR as a predictor of the outcome of COVID-19 patients. The primary endpoint was mortality, while the second endpoint was admission to an intensive care unit. For this purpose, a comparison of the main biomarkers (CRP, NLR, PLR) was also performed.

We also evaluated the prognostic role of P/F at baseline, which is the standard parameter assessing pulmonary function, being the ratio of partial arterial O_2_ pressure (PaO_2_) over the fraction of inspired O_2_ (FiO_2_).

## 2. Methods

### 2.1. Subjects

In this retrospective two-center observational survey, 411 subjects were selected among 512 patients who were consecutively admitted to two hospitals in Catania, Italy, with a reliable diagnosis of SARS-CoV2 obtained by rRT-PCR from September 2020 to May 2021. A selection of patients was then made according to the inclusion criteria listed below:No past or present medical history of chronic illness affecting leukocyte formula (i.e., autoimmune diseases, malignancies, inflammatory chronic diseases);No previous chronic pharmacological treatment affecting leukocyte formula before admission to the hospital;Availability of at least two complete blood counts and blood gas tests;Hospitalization > 48 h.

Thus, patients on chronic therapy for hematological, neoplastic, and inflammatory comorbidities were not allowed to enter the present survey (Figure 1).

All data were obtained from clinical records, and information was adequately de-identified before statistical analysis. Blood samples for routine biohumoral parameters (including total and differential leukocyte counts, platelet counts, and serum levels of C-Reactive Protein (CRP) were collected at baseline, on the median day of hospitalization, and at discharge (to ICU or another ward or home) and assayed by autoanalyzers (Beckman Coulter DxH 800; Danaher Corporation, Miami, FL, USA, Beckman DxC 700 AU; Danaher Corporation, Miami, FL, USA).

Regarding the calculation of ratios of peripheral blood cells, the following formulae were applied:

- NLR: number of neutrophils/number of lymphocytes;

- PLR: number of platelets/number of lymphocytes.

The derived neutrophil-to-lymphocyte ratio (d-NLR) was also calculated using the following formula: number of neutrophils/(total leukocyte − countnumber of neutrophils). The P/F ratio was also obtained from hemogasanalysis as the ratio of PaO_2_/FiO_2_ (Table 1).

According to the primary endpoint of the study, the number of deaths due to COVID-19 infection was also recorded.

### 2.2. Statistical Analysis

Statistical analysis was performed using the Medcalc and IBM-SPSS packages.

Categorical variables were described in terms of absolute frequency and percentage, whereas the continuous variables were further subdivided into two groups, based on the result of the Kolmogorov–Smirnov test and expressed as a mean ± standard deviation in the case of normally distributed variables, whereas as median and relative interquartile range in case of non-gaussian variables.

Where appropriate, variables characterized by extreme asymmetry or kurtosis were normalized by z score.

The study population was divided into tertiles according to the NLR range, and the differences were assessed using the Chi-square test for categorical variables, one-way ANOVA for normally distributed continuous variables, and Kruskal–Wallis test for non-gaussian continuous variables.

In order to derive the best cut-offs of the different markers, to test their specificity and sensitivity, and to compare the area under the curve, multiple ROC curves were also plotted.

First, the relationship of NLR expressed as a continuous variable was related to in-hospital mortality and admission to ICU; this relationship was then analyzed considering NLR as a variable categorized into tertiles.

Univariate and multivariate logistic regression models were built to evaluate these relationships. Adjusted and unadjusted odds ratios (ORs), as well as 95% confidence intervals (CI), were also calculated.

A survival analysis using a Kaplan–Meier curve was performed, followed by a Cox proportional hazards regression analysis. The null hypothesis was excluded, in all two-tailed tests, for *p* values < 0.05.

## 3. Results

### Steroid Treatment

During hospitalization, all patients were given a mean dose of 20 mg of steroids once a day.

The main clinical characteristics of the overall study population are summarized in Table 2.

The median age was 72 years (interquartile range: 70–75). It increased significantly from the first to the third tertile. A higher prevalence of male sex was observed, but no significant difference was found between subgroups. As to hematochemical data (Table 2), a statistically significant increase in WBC (*p* = 0.000009), CRP (*p* < 0.000001), PLR (*p* < 0.000001), and D-NLR (*p* < 0.000001) was observed from the first to the third tertile.

With regard to differential WBC count, neutrophils showed a significant increase along tertiles, while the lymphocytes count exhibited an opposite trend; all patients, except those belonging to the first tertile, showed a neutrophilic leukocytosis with lymphopenia. Compared to the first tertile, no difference was found in the distribution of platelets count in the upper two tertiles.

No difference was also observed between the tertiles regarding the main comorbidities. A first ROC curve built for NLR expressed as a continuous variable showed an acceptable predictive power of intra-hospital mortality, namely the first endpoint (AUC = 0.772 *p* < 0.0001): the best cut-off of NLR established by the Youden index at 11.38 had a sensitivity of 72.9% and a specificity of 71.9%. A comparison between ROC curves was made in order to test the prognostic performance of NLR, D-NLR, PLR, and CRP (Figure 2) for predicting in-hospital mortality of COVID-19 patients. NLR showed the largest area under the curve, followed by CRP and PLR. The difference between NLR and CRP AUCs (*p* = 0.0173), as well as that between NLR and PLR (*p* < 0.0001), was statistically significant. The number of deaths, as well as the number of ICU admissions, significantly increased from the first to the third tertile. The median length of stay did not show a significant difference in the subgroups (Figure 3).

Survival analysis using Kaplan–Meier curves was performed (Figure 3). A statistically significant (*p* < 0.0001) difference in survival among patients throughout NLR tertiles was also observed.

Two Cox regression models were built in order to obtain Hazard Ratios of the main markers in each NLR tertile, adjusted for the main confounders (Table 3). The model was adjusted for sex and age, while model 2 was for the main comorbidities and for the P/F ratio. NLR was independently associated with mortality. In model 1, for each increase of 1 unit in the standard deviation of NLR, the risk of mortality increased by 45% in the whole sample. In model 2, for each increase of 1 unit in the standard deviation of NLR the risk of mortality increased by 60% in the whole sample. As for CRP, in model 1, as well as in model 2, for an increase of 1 unit in the standard deviation of CRP, the risk of mortality increased by 4% in the whole population.

The lymphocyte count seems to be protective: for a reduction of 100 units of lymphocytes, the risk of mortality increased by 3% in model 1 and by 4.5% in model 2 (Table 3). As for the neutrophil count, for each increase of 1000 units of neutrophils, the risk of mortality increased by 10% and 75% in the first and second models, respectively, after adjustment for confounders. An increase in mortality risk of 3% was observed with an increase of each unit of D-NLR in both Cox models.

For the secondary endpoint, 79 patients (19%) entered the ICU. The Youden index identified a cut-off of NLR of 8.21 as the optimal predictor of admission to ICU. This cut-off is characterized by an area under the curve of 0.664, with acceptable sensitivity (81%) but low specificity (49%). Comparisons of the ROC curves related to NLR, CRP, and PLR showed no significant differences (Figure 4).

Moreover, Table 4 shows Cox regression models regarding the prediction of ICU admission.

Cox regression analysis showed that NLR largely predicted the risk of ICU admission (HR: 3.9597, *p* < 0.0001). The independent association of NLR with ICU admission, however, disappeared when P/F was included in the model. This means that NLR and P/F are associated (Figure 5) in the same pathophysiological chain, where the weight of NLR is entirely captured by P/F, likely because NLR behavior is linked to the worsening P/F, which, in turn, characterizes hypoxemic respiratory failure.

Covariate selection was determined by a gradual process, excluding all extremely collinear covariates with a high Variance Inflation Factor, such as fibrinogen, neutrophil count, and others. To understand whether coagulation markers were involved in predicting prognosis, logistic regression models including the available coagulation markers (D-dimer, fibrinogen, and INR) were built, but no significant influence of these coagulation markers was shown.

Finally, we attempted to evaluate the behavior of NLR and CRP, throughout hospitalization. NLR and CRP were recorded at three-time intervals, specifically the first, last and median days of hospitalization (Figure 5). Both NLR and CRP are elevated at the beginning of COVID-19 disease; while the median levels of CRP showed a decrease (−19.8) throughout hospitalization, NLR tended to increase (+3.5), thus depicting an asynchronous pattern, where CRP tends to fall rapidly, while NLR tends to further increase more slowly. Moreover, in deceased subjects, CRP and NLR showed the same asynchronous trend (CRP: −13.20; NLR: +8.8), with a more pronounced later increase in NLR (Figure 6b).

## 4. Discussion

To the best of our knowledge, our study demonstrates for the first time that patients with higher NLR showed a higher risk of intra-hospital mortality and disease progression to a stage of severity that requires admission to ICU. Of note, the prediction of mortality by Cox models (Table 2) was much larger for NLR than for CRP. Moreover, comorbidities were not associated with a worse prognosis. This could at first-hand look surprising because the literature has identified several chronic diseases as major risk factors associated with mortality and disease progression [13] in COVID-19 patients. A recently published WHO Report [14] identified hypertension as a powerful risk factor for COVID-19 infection, hospitalization in ICU, severe illness and mortality. However, the weakened role of comorbidities observed in our study could depend on the large prevalence of comorbidities in our geriatric population so that their impact on prognosis was so widespread that their individual influence on outcome may result smoothened.

It is well known that CRP is a recognized prognostic inflammatory biomarker that plays an important role in pathogen resistance and immune response. It is consistently correlated with an adverse outcome and mortality in numerous studies, as well as with acute lung injury [5,15]. Furthermore, elevated serum CRP levels have been repeatedly associated with the onset of cardiovascular events, the development of acute respiratory distress syndrome (ARDS), and mortality due to infection by SARS-CoV-2 [16,17]. Our study confirmed an association between CRP levels and mortality and/or adverse outcome.

NLR is believed to reflect the balance between innate and adaptive immune responses [18]. It also appears to be a better tool than the absolute neutrophils and lymphocytes counts, being less affected by the effect of various confounders. Moreover, NLR has been previously recognized as a diagnostic tool and predictive marker of disease severity in patients with influenza virus and other inflammatory diseases [19]. Some studies have also identified the role of NLR in discriminating severe diseases and forecasting mortality. Recently, Citu et al. [20] and Kheyri et al. [21], in two single-center studies, demonstrated the importance of NLR in predicting death or ICU admission in COVID-19 patients, although serial NLR and CRP data, in contrast to our study, were not recorded. Moreover, growing evidence supports the role of neutrophils in SARS-CoV-2 infection. The survival of neutrophils seems in fact to be prolonged for several days following the viral infection. This prolonged activation leads in turn to the release of pro-inflammatory mediators and toxins, paving the way for the need for repeated measurements of NLR throughout hospitalization, as conducted in our study. There was also evidence that neutrophils chair nonspecific immunity that starts the body’s responses to inflammation, while lymphocytes are protective elements against inflammation, being important for dampening innate immune responses during viral infection [22]. Furthermore, plasma levels of cortisol and catecholamines are the main endogenous stimuli for an increase in NLR due to both neutrophilia and lymphopenia; cytokines and other hormones are likely to run in the same direction as well [23]. In cases of viral and bacterial pneumonia, NLR was also an effective biomarker of the severity of CAP [6,24].

In our population, the NLR increase could be suggestive of the inflammatory response to COVID-19, somehow reflecting the imbalance between inflammation and immune response. For each increase of 1 unit in NLR occurred an increase in mortality ranging from 45% (Cox model 1 Table 3) to 59% (Cox model 2, Table 3). Mortality was also predicted by CRP, ANC, and D-NLR but not by PLR (Table 3). Some studies have suggested that lymphopenia indicates that SARS-CoV-2 has a cytotoxic effect on B lymphocytes, associated with a peripheral hyperactivation of T lymphocytes, both phenomena influencing the prognosis of these patients [17,25]. Our results are in agreement with data by Tan et al. [25], emphasizing the usefulness of monitoring NLR to identify the severity of prognosis in COVID-19 patients, as shown in Figure 3.

The inflammation-mediated effect of COVID-19 on lymphocytes was also observed in autoptic material, showing early and diffuse alveolar lesions with exudate, paving the way to the reduction in lymphocyte count [26] previously observed in Severe Acute Respiratory Syndrome (SARS) and Middle East Respiratory Syndrome (MERS) [27]. This concept is also corroborated by our data emphasizing that NLR is independently associated with ICU admission, which is almost three times higher in parallel with NLR increase. However, this prediction is abolished when is corrected for P/F values, meaning that both NLR and P/F are involved in the same pathogenetic chain, where the impact of inflammation (i.e., NLR changes) is statistically captured by the consequent functional worsening in P/F, characterizing respiratory failure (Figure 5).

As a matter of fact, lymphocytopenia was also recently described in association with the disease’s severity and cytokine storm in COVID-19 patients by Terpos et al. [28]. Several studies have highlighted the prognostic role of PLR to predict mortality and disease severity, as well as its correlation with the period of hospitalization [29,30]. In our study, PLR, despite a statistically significant area under the ROC curve and acceptable sensitivity and specificity (Figure 2), failed to predict mortality (Table 3), although its ROC curve resulted similarly to those of NLR and CRP with regard to ICU admission (Figure 4). Our results on PLR in COVID-19 patients are in agreement with previous data [31,32,33], emphasizing no association between PLR and mortality. Other studies showing an association of PLR with mortality included patients with thrombocytopenia, which could be a confounder [34,35]. On the whole, a recent metanalysis on the role of PLR as a prognostic biomarker in COVID-19 patients is heterogeneous [36], not allowing to consider PLR a useful marker of outcome in this setting.

The D-NLR also showed an acceptable prognostic value, less significant than NLR (Figure 2), so that its predictive value, as already known, would remain uncertain [37].

Finally, the asynchronous pattern of NLR and CRP (Figure 6) would emphasize that NLR and CRP are both elevated at the beginning of the COVID-19 disease, but NLR further increased later, particularly in deceased patients faced with a fall in CRP. This would mean that NLR provides additional information on the prolonged inflammatory state in COVID-19 patients, particularly in those with a worse prognosis. Panel C in Figure 6 showed that NLR and CRP have a similar behavior before admission to ICU, emphasizing that both markers run in parallel. This would mean that at this stage, there is no difference between NLR and CRP in the ability to identify the degree of severity.

In conclusion, NLR represents a rapid, widely available and relatively inexpensive tool that could be useful in the management and early risk stratification of patients with COVID-19. Our study provided additional evidence to recent data by Vuillaume et al. [38] obtained in patients followed-up either in the ward or as out-patients. At variance, our data were obtained only in hospitalized COVID-19 patients, in whom a panel of biomarkers were compared with NLR, thereby emphasizing its prominent prognostic role, as well as both the pattern of its increase over time compared with CRP and the inverse relationship with P/F.

To the best of our knowledge, our study is therefore the first to demonstrate, in a retrospective study involving two different centers, the prognostic role of NLR compared with other biomarkers in a selected cohort of hospitalized COVID-19 patients free of confounders affecting the leukocyte count. Our study has some strengths: the ability to demonstrate (a) the predictive prognostic role of NLR, independently of comorbidities, (b) the inverse relationship of NLR with P/F, and (c) the different patterns of NLR as compared with CRP throughout hospitalization. It has also limitations: It is a retrospective study carried out on patients with homogeneous demographic and clinical characteristics. Further prospective multicentric studies are needed, which are also finalized to identify standardized cut-off values of NLR, to be used worldwide to stratify the disease’s severity in patients with COVID-19 infection.

## Figures and Tables

**Figure 1 jcm-11-02235-f001:**
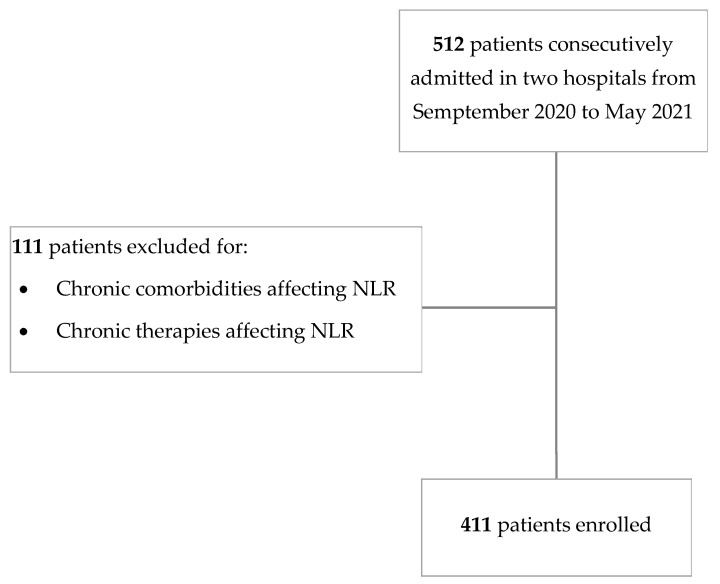
Flow chart of patient selection. NLR = Neutrophil-to-lymphocyte ratio.

**Figure 2 jcm-11-02235-f002:**
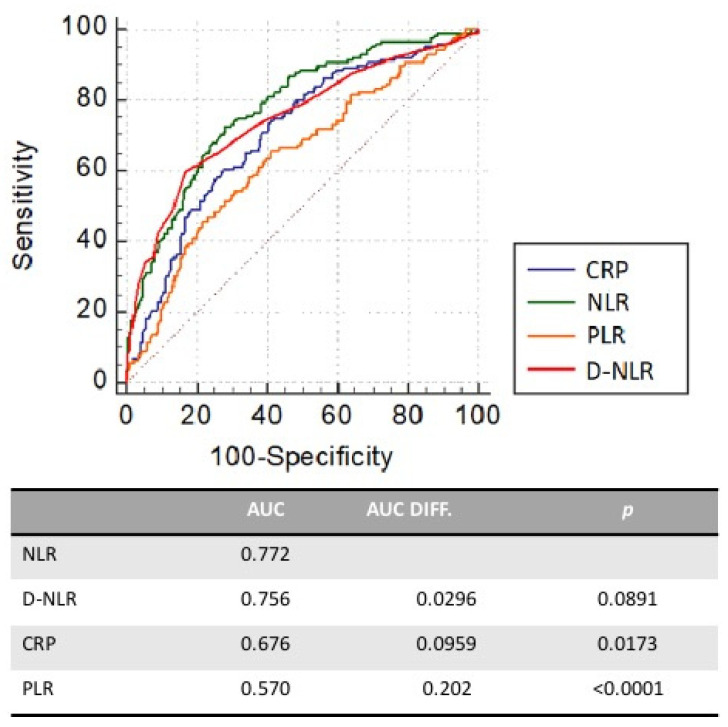
Receiver operating characteristic (ROC) curve of NLR, d-NLR, CRP, and PLR for predicting in-hospital mortality of COVID-19 patients.

**Figure 3 jcm-11-02235-f003:**
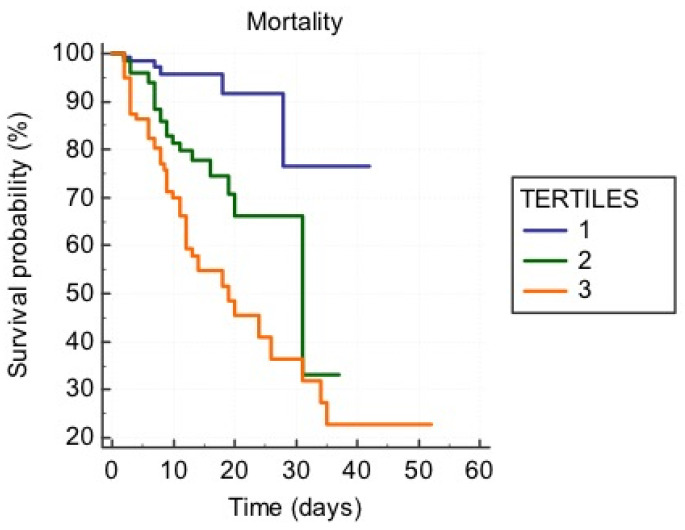
Kaplan–Meier curve showing survival probability according to NLR tertiles. (1st NLR tertile = 0.9–6.83; 2nd NLR tertile = 6.89–14.02; 3rd NLR tertile = 14.1–98).

**Figure 4 jcm-11-02235-f004:**
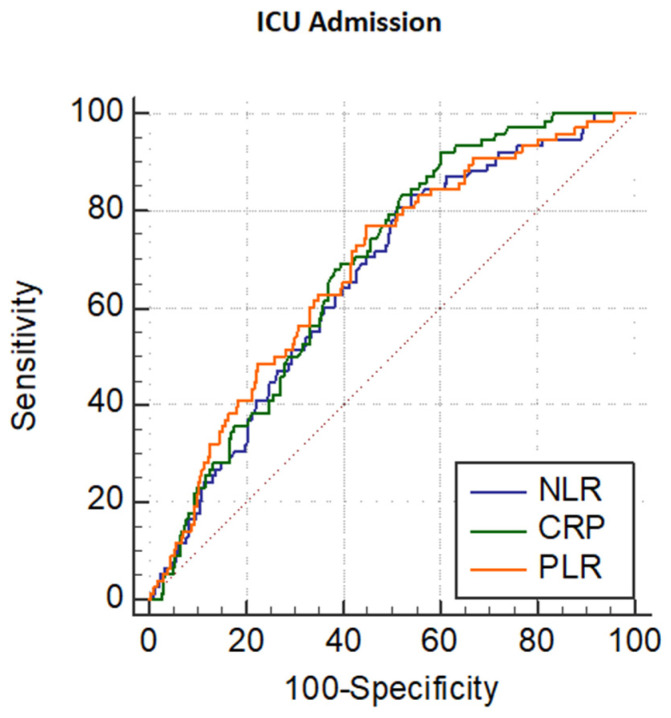
Receiver operating characteristic (ROC) curve of NLR, CRP, and PLR for predicting admission to ICU of patients with COVID-19.

**Figure 5 jcm-11-02235-f005:**
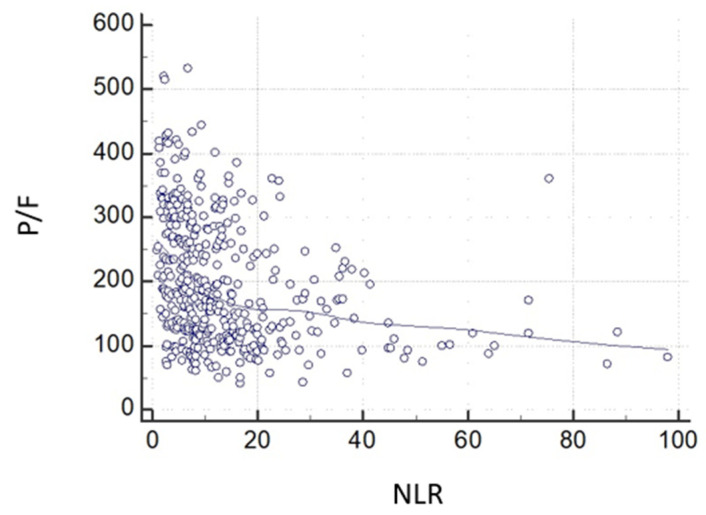
Relationship between the neutrophil-to-lymphocyte ratio and the P/F ratio at baseline (r = 0.39; *p* < 0.0001).

**Figure 6 jcm-11-02235-f006:**
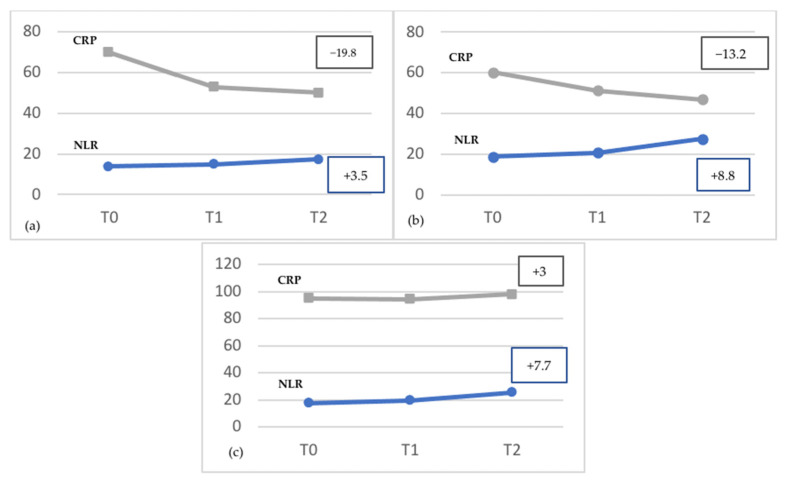
Time course of mean values of NLR and CRP in the total sample (**a**), in deceased patients (**b**), and in patients admitted to ICU (**c**). (T0 = admission to hospital; T1 = median day of hospitalization; T2 = last day of hospitalization).

**Table 1 jcm-11-02235-t001:** Baseline flow rate, FiO_2_, and PaO_2_/FiO_2_ of oxygen supplementation.

	Low Flow Oxygen Therapy			High Flow Oxygen Therapy		NIMV			
	Nasal Cannula		Venturi Mask	Venturi Mask	HFNC	C-PAP			NIV
	1–3 L/min (FiO_2_ = 0.24–0.32)	4–6 L/min (FiO_2_ = 0.36–0.44)	4–8 L/min (FiO_2_ = 0.24–0.35)	10–12 L/min (FiO_2_ = 0.40–0.60)	FiO_2_ = 0.40–0.60	FiO_2_ <= 0.40	FiO_2_ = 0.41–0.50	FiO_2_ = 0.51–0.60	
Patients, *n*	29	26	65	39	47	16	58	30	32
PaO_2_/FiO_2_	237.8 (193.8–281.2)			162.43 (121–218.8)		119.9 (92.5–151) *			

* *p* < 0.000001 referred to the difference among all PaO_2_/FiO_2_ values. HFCN = High Flow Nasal Cannula; NIMV = Non-Invasive Mechanic Ventilation; C-PAP = Continuous-Positive Air Pressure; NIV = Non-Invasive Ventilation.

**Table 2 jcm-11-02235-t002:** Baseline demographic, laboratory, and clinical characteristics of the sample and features of different subgroups based on NLR tertiles. Reported in bold is the significance between tertiles. * Significant difference from the first tertile.

	Total (*n* = 411)	1st Tertile (*n* = 137)	2nd Tertile (*n* = 137)	3rd Tertile (*n* = 137)	*p*
**Age, y**	72 (70–75)	67 (61–69)	71 (68–74)	80 (77–83)	**<0.000001**
**Male sex, *n* (%)**	237 (57.7)	71 (51.9)	88 (64.3)	78 (57)	0.1127
**P/F**	173 (161–187)	244 (215- 269.6)	163 (51–184)	129 (120–145)	**<0.000001**
**WBC, 10^9^/L**	8.8 (8.2–9.4)	6.1 (5.8– 6.9)	8.8 (8.4–9.7)	11.7 (10.8–12.9)	**0.000009**
**ANC, 10^9^/L**	7.2 (6.8–7.9)	4.4 (4.1- 5)	7.5 (7.1–8.2)	10.5 (9.8–11.7)	**<0.000001**
**LYMPH, 10^9^/L**	0.73 (0.4–0.8)	1.20 (1–1.3)	0.76 (0.7–0.8)	0.48 (0.4–0.5)	**<0.000001**
**PLT, 10^9^/L**	226 (217–235.8)	215 (194.3–234.4)	237 (221–254)	222 (210–242.9)	0.0553
**CRP, mg/dL**	8.9 (8.13–9.59)	4.9 (3.7–5.9)	9.7 (8.4–11.2)	13 (10.6–15)	**<0.000001**
**PLR**	226 (217–235.8)	215 (194.3–234.4)	237 (221–254)	222 (210–242.9)	**<0.000001**
**d-NLR**	5 (5–6)	3 (2–3)	6 (5–6)	10.5 (9–12)	**<0.000001**
**Length of stay, days**	10 (9–11)	8 (7–10)	10 (8–12)	11 (9–12)	0.1444
**Deceased, *n* (%)**	133 (19.5)	13 (4.4)	39 (16.8)	81 (37.3)	**<0.00001**
**ICU Admission, *n* (%)**	79 (19.3)	10 (7.3)	30 (21.9)	39 (28.5)	**0.000032 ***
**Hypertension, *n* (%)**	244 (59.4)	79 (51.9)	76 (55.5)	89 (65)	0.2461
**Diabetes Mellitus, *n* (%)**	111 (27)	32 (23.4)	37 (23.4)	42 (30.7)	0.3962
**CKD, *n* (%)**	35 (8.6)	10 (7.3)	9 (6.6)	16 (11.7)	0.2610
**COPD, *n* (%)**	34 (8.3)	11 (8.1)	11 (8.1)	12 (8.8)	0.9684
**CV Disease, *n* (%)**	70 (17.1)	22 (16.1)	18 (13.2)	30 (21.9)	0.1453

P/F = PaO_2_/ FiO_2_; WBC = white blood cells; ANC = absolute neutrophil count; LYMP = lymphocytes; PLT = platelets; CRP = C-reactive protein; PLR = Platelet-to-Lymphocyte Ratio; d-NLR = derived neutrophil-to-lymphocyte ratio; ICU = Intensive Care Unit; CKD = chronic kidney disease; COPD = chronic obstructive pulmonary disease; CV = cardiovascular; n = number; y = years.

**Table 3 jcm-11-02235-t003:** Cox regression models predicting mortality. Model 1: adjusted for sex and age. Model 2: adjusted for comorbidities.

NLR	Unadjusted HR	*p*	Model 1 HR	*p*	Model 2 HR	*p*
**1° tertile**	1.20	0.4902	1.20	0.4821	1.16	0.5952
**2° tertile**	1.25	**0.0296**	1.28	0.0267	1.26	**0.0314**
**3° tertile**	1.38	**0.0009**	1.29	0.0108	1.38	**0.0018**
**Total**	1.62	**<0.0001**	1.45	<0.0001	1.59	**<0.0001**
	**Unadjusted HR**	** *p* **	**Model 1 HR**	** *p* **	**Model 2 HR**	** *p* **
**CRP**	1.04	**<0.0001**	1.04	<0.0001	1.04	**<0.0001**
**ANC**	1.17	**<0.0001**	1.10	0.0259	1.75	**0.0001**
**LYMP**	0.5965	**0.0089**	0.7030	0.0437	0.5501	**0.0114**
**d-NLR**	1.05	**<0.0001**	1.03	0.0010	1.03	**0.0106**
**PLR**	1	0.6800	1	0.6605	1	0.6958

HR = hazard ratio; NLR = neutrophil-to-lymphocyte ratio; CRP = C-reactive protein; ANC = absolute neutrophil count; LYMP = lymphocytes; d-NLR = derived neutrophil-to-lymphocyte ratio; PLR = platelet-to-lymphocyte ratio.

**Table 4 jcm-11-02235-t004:** Cox regression models predicting ICU admission. Model 1: adjusted for sex and age. Model 2: adjusted for comorbidities. Model 3: adjusted for P/F.

	Unadjusted HR	*p*	Model 1 HR	*p*	Model 2 HR	*p*	Model 3 HR	*p*
NLR	3.9597	**<0.0001**	1.02	**0.0035**	1.01	**0.0391**	1.005	0.4741

HR = hazard ratio; NLR = neutrophil-to-lymphocyte ratio.

## Data Availability

The data presented in this study are available on request from the corresponding author.

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
