# Peer review of "Neutrophil-to-Lymphocyte Ratio (NLR) Is a Promising Predictor of Mortality and Admission to Intensive Care Unit of COVID-19 Patients"

_jcm, 2022, doi:10.3390/jcm11082235_

Round 1

Reviewer 1 Report

Regolo et al. ‘s article aims to evaluate the predictive role of Neutrophil-to-Lymphocyte ratio (NLR) of the mortality and admission in intensive care unit in Covid-19 patients. 411 patients were retrospectively selected in two hospitals. Three inflammatory biomarkers were compared in those patients: C-Reactive Protein (CRP), Neutrophil to Lymphocyte Ratio (NLR) and Platelet to Lymphocyte Ratio (PLR). Based on statistical analysis, the authors provide evidence that NLR is an independent predictor of mortality in Covid-19 patients.

The article is of interest. As described by the authors, the NLR is indeed an interesting biomarker of inflammation. Its study in the framework of Covid-19 and based on the literature, is therefore relevant, especially in hospitalized patients. However, a major concern is regarding the inclusion criteria. Patients were including if “no previous pharmacological treatment affecting leukocyte formula”. Since the second Covid-19 wave, patients are mainly treated with corticoids which affect leukocyte production. This article will be even more relevant with a section detailing the patients’ treatments. In the same direction, the article evaluates the prognostic role of P/F ratio but no data concerning oxygen therapy and in particular the flow rate is indicated in the methods. Adding these information will increase the relevance of the results presented in this article. 

Lines 62 to 66: Paragraph would be stronger with references added.

Line 94: Methods; the name of the autoanalyser for routine blood samples would improve the methods.

Table 1: P/F data are not described

Line 203: Cox regression analysis predicting ICU admission is not showed.

Figure 5 will benefit to have the time course of NLR and CRP in the patients admitted to the ICU.

Line 317: Number of Clinical Trials Declaration is not declared

Author Response

Response to Reviewer 1 Comments

Point 1: However, a major concern is regarding the inclusion criteria. Patients were including if “no previous pharmacological treatment affecting leukocyte formula”. Since the second Covid-19 wave, patients are mainly treated with corticoids which affect leukocyte production. This article will be even more relevant with a section detailing the patients’ treatments.

Response 1: We thank the Reviewer for the comment. In figure 1 (Line 100) it is specified that patients on chronic treatment for comorbidities and on chronic therapies affecting NLR were excluded from this survey. Moreover during hospitalization all patients included in this survey were given a mean dose of 20 mg of steroids (Line 172).

Point 2: In the same direction, the article evaluates the prognostic role of P/F ratio but no data concerning oxygen therapy and in particular the flow rate is indicated in the methods.

Response 2: We thank the Reviewer for the comment. A new table (Table 1) was added to specify different P/F values according to oxygen supplementation. (Line 116)

Point 3: Lines 62 to 66: Paragraph would be stronger with references added.

Response 3: We thank the Reviewer for the comment. References were added accordingly. (Line 63-67)

Point 4: Line 94: Methods; the name of the autoanalyser for routine blood samples would improve the methods.

Response 4: We thank the Reviewer for the comment. Name of the autoanalysers was added. (Line 108)

Point 5: Table 1: P/F data are not described

Response 5: We thank the Reviewer for the comment. As we indicated in “Point 2”, data concerning P/F were added. (Line 116)

Point 6: Line 203: Cox regression analysis predicting ICU admission is not showed.

Response 6: We thank the Reviewer for the comment. Table 4 was added to describe Cox regression analysis regarding patients admitted to ICU. (Line 262)

Point 7: Figure 5 will benefit to have the time course of NLR and CRP in the patients admitted to the ICU.

Response 7: We thank the Reviewer for the comment. A new panel on NLR and CRP in patients admitted to ICU was added in Figure 6. (Line 293)

Point 8: Line 317: Number of Clinical Trials Declaration is not declared.

Response 8: We thank the Reviewer for the comment. There is no Identification Numer of the Trial but only the specific aim, as stated in the acknowledgments. (Line 467)

Reviewer 2 Report

It is an interesting and well-written article. English seems to be correct however, I cannot be considered as English native. In this study Regolo and colleagues documented the sensitivity and specifity of neutrophil-to-lymphocyte ratio in prediction of mortality and admission to ICU in patients treated for COVID-19. Despite the promising findings I have one comment.

Introduction. Whole introduction describes a problem of NLR. However the authors also studied PLR and CRP. Therefore they should write some sentence describing a usefulness of PLR and CRP in predicting mortality in COVID-19 patients. The authors did not find correlation between PLR and mortality rate while several studies documented its usefulness in COVID-19 patients. This difference required extensive discussion, not only one short paragraph.   

The authors should add some sentence described findings with PLR.

References. The format of references have to be changed in accordance to journal guidelines.

Author Response

Response to Reviewer 2 Comments

Point 1: Introduction. Whole introduction describes a problem of NLR. However the authors also studied PLR and CRP. Therefore they should write some sentence describing a usefulness of PLR and CRP in predicting mortality in COVID-19 patients. The authors did not find correlation between PLR and mortality rate while several studies documented its usefulness in COVID-19 patients. This difference required extensive discussion, not only one short paragraph. The authors should add some sentence described findings with PLR.

Response 1: We thank the Reviewer for the comment. We added the comment to the discussion regarding PLR. (Line 376-381)

Point 2: References. The format of references have to be changed in accordance to journal guidelines.

Response 2: We thank the Reviewer for the comment. Reference list was modified in accordance to journal guidelines.

Round 2

Reviewer 1 Report

The authors adressed all my comments . One correction ligne 193, fig2 is now figure3. 

Author Response

We thank the Reviewer for the comment. The numbering of figure 3 was modified.

This manuscript is a resubmission of an earlier submission. The following is a list of the peer review reports and author responses from that submission.